



# Size Resolved Online Chemical Analysis of Nano Aerosol Particles: A Thermal Desorption Differential Mobility Analyzer Coupled to a Chemical Ionization Time Of Flight Mass Spectrometer

Andrea C. Wagner[1], Anton Bergen[1], Sophia Brilke[1,2], Claudia Fuchs[1,3], Markus Ernst[1], Jesica Hoker[1], Martin Heinritzi[1], Mario Simon[1], Bertram Bühner[1], Joachim Curtius[1], Andreas Kürten[1]

[1]Institute for Atmospheric and Environmental Sciences, Goethe University Frankfurt, Frankfurt, 60438, Germany
[2]Aerosol Physics and Environmental Physics, University of Vienna, Vienna, 1090, Austria
[3]Laboratory of Atmospheric Chemistry, Paul Scherrer Institute, Villigen, 5232, Switzerland

*Correspondence to*: Andrea C. Wagner (acwagner@iau.uni-frankfurt.de)



**Abstract.** A new method for size resolved chemical analysis of nucleation mode aerosol particles (size range from ~10 to ~30 nm) is presented. The Thermal Desorption Differential Mobility Analyzer (TD-DMA) uses an online, discontinuous principle. The particles are charged, a specific size is selected by differential mobility analysis and they are collected on a filament by electrostatic precipitation. Subsequently, the sampled mass is evaporated in a clean carrier gas and analyzed by a chemical ionization mass spectrometer. Gas phase measurements are performed with the same mass spectrometer during the sampling of particles. The characterization shows reproducible results, with a particle size resolution of 1.19 and the transmission efficiency for 15 nm particles being slightly above 50 %. The signal from the evaporation of a test substance can be detected starting from 0.01 ng and shows a linear response in the mass spectrometer. Instrument operation in the range of pg/m$^3$ is demonstrated by an example measurement of 15 nm particles produced by nucleation from dimethylamine, sulfuric acid and water.

# 1 Introduction

Aerosol particles play an important role in the earth's climate. They influence the radiative budget directly by scattering and absorbing solar radiation, and indirectly by changing cloud properties such as albedo and lifetime. In climate models, these interactions are still the largest source of uncertainty (Core Writing Team et al., 2007; Fuzzi et al., 2015). Aerosol particles also affect air quality and human health. Depending on their size and shape, they can protrude deep into the lungs (Kreyling et al., 2006) and even enter the bloodstream (Nel, 2005), causing problems such as ischemic stroke (Wellenius et al., 2012), premature mortality (Lelieveld et al., 2015) and many others (e.g. Pope III and Dockery, 2006; Donaldson and Borm, 2006).

A large fraction of atmospheric aerosol particles originates from new particle formation (Merikanto et al., 2009; Dunne et al., 2016). Several major formation mechanisms have already been subject of intense research (Hallquist et al., 2009; Zhang et al., 2012, Kulmala et al., 2014; Kirkby et al., 2016), yet various reaction pathways and processes remain unknown. The newly formed particles need to grow to a certain size (~ 50-100 nm) to act as cloud condensation nuclei (e.g. Hallquist et al., 2009; Riccobono et al., 2012; Vehkamäki and Riipinen, 2012; Tröstl et al., 2016; Lehtipalo et al., 2016). In order to better understand the nucleation and subsequent growth to cloud condensation nuclei (CCN), the condensing vapors, the freshly nucleated small particles, as well as the larger particles need to be chemically speciated and quantified.

For the analysis of atmospherically relevant nucleation precursors, chemical ionization mass spectrometry is quite commonly used. Depending on the target analyte, the primary ion is chosen. Prominent examples include the proton transfer reaction technique using hydronium ions (Hansel et al., 1995; Graus et al., 2010; Breitenlechner et al., 2017), successfully targeting volatile organic compounds, or negatively charged nitrate ions for the detection of sulfuric acid (Eisele and Tanner,



1993; Kürten et al., 2011; Jokinen et al., 2012), the class of extremely low volatile organic compounds (ELVOCs) (Ehn et al., 2014), and clusters of sulfuric acid and dimethylamine (Kürten et al., 2014).

Particles that have already grown to larger sizes are frequently analyzed by the well-established technologies of the aerosol mass spectrometer (AMS) (Jayne et al., 2000; Zhang et al., 2011) and single particle mass spectrometry (Noble and Prather, 2000; Bzdek et al., 2012). The chemical composition of particles, however, changes with size, and obtaining detailed chemical information about smaller particles is very challenging as they have an extremely low mass.

To close the gap between the measurement of gas phase and larger particles, a number of interesting techniques emerged in the past few years. Only few instruments are capable of analyzing sub 30 nm particles. They can be distinguished based on the following main criteria (see Table 1): (1) a discontinuous principle to enrich the analyte vs continuous measurements, (2) size resolved methods sampling by electrical precipitation vs integral sampling methods, (3) evaporation of particles by thermal desorption vs laser evaporation, and (4) ability to analyze gas and particle phase vs only particle phase.

The nanoaerosol mass spectrometer (NAMS) (Wang et al., 2006) is a single particle mass spectrometer for small particles and uses a continuous principle. It consists of an aerodynamic lens, ion guide and quadrupole ion trap with laser evaporization. The Volatile Aerosol Component Analyzer (VACA) (Curtius et al., 1998; Curtius and Arnold, 2001) is able to continuously measure sulfuric acid in sub 30 nm particles in aircraft plumes, where gas phase concentrations are much lower than particle phase concentrations. The thermal desorption chemical ionization mass spectrometer (TDCIMS) (Voisin et al., 2003) is a size resolved method collecting particles electrostatically and using thermal desorption to measure the particle phase composition. It has provided important insights to several chemical systems including aerosol from marine environments (Lawler et al., 2014), and has even been able to perform size resolved measurements for extremely small particle sizes of 8-10 nm (Smith et al., 2010). The aerosol mass spectrometer by Laitinen et al. (2009) precipitates particles on a platinum surface and emits them by laser ablation. It has been used for organic compounds in the size range of 10 to 50 nm (Laitinen et al., 2011). Two other size selective techniques using thermal desorption are the devices by Phares and Collier (2010) and the CaCHUP by Gonser and Held (2013). The Filter Inlet for Gases and AEROsols (FIGAERO) (Lopez-Hilfiker et al., 2014) is a bulk phase filter sampler using thermal desorption. Although not providing size resolved information, it is able to measure both gas and particle phase and thus to investigate partitioning effects (Lopez-Hilfiker et al., 2015).

In this study, we present our novel device for the online chemical analysis of nucleation and Aitken mode particles. The Thermal Desorption - Differential Mobility Analyzer (TD-DMA) is capable of measuring in a size resolved and integral setting. It allows taking gas phase measurements with the same mass spectrometer. As a mobile interface, it can be combined with different mass spectrometers or other real time gas phase analyzers.

Here, the instrument is described (Sect. 2), followed by a detailed characterization (Sect. 3) comprising the DMA unit's transmission efficiency, the particle collection efficiency, the filament temperature and the reproducibility of the evaporation process. The measurement procedure (Sect. 4) is specified, including determination of the background and an



exemplary measurement of 15 nm particles. Lastly, the signal is quantified (Sect. 5) and the results are discussed (Sect. 6), closing with a short summary (Sect. 7).

## 2 Instrument Description

The TD-DMA is designed for size resolved chemical analysis of nanometer sized aerosol particles. The particles are charged, a specific size is selected and they are electrostatically collected on a filament. Subsequently, the sampled mass is evaporated in a clean carrier gas to be analyzed by a detector, e.g. a mass spectrometer. In this way, the particle phase is efficiently separated from the gas phase and the concentration is enhanced to meet the detection limit of the mass spectrometer. The coupling between TD-DMA and detector allows for gas phase measurements during particle sample collection periods. This enables obtaining a broad picture of both particle and gas phase chemical composition with the same detector and to observe the partitioning of the analyzed substances.

The TD-DMA is modular in its setup as it can in principle be combined with any detector suitable for real-time chemical analysis of gas phase compounds. In this case, individual compounds relevant for nucleation and early growth of atmospheric nano aerosol particles should be measured. Therefor we used a chemical ionization atmospheric pressure interface time-of-flight (CI-APi-TOF) mass spectrometer with negative nitrate primary ions generated by a corona ion source (Kürten et al., 2011). This technique is specialized for the detection of sulfuric acid, amines and highly oxigenated organic molecules (HOMs) (Kürten et al., 2014; Simon et al., 2016; Kirkby et al., 2016) and characterized for its internal transmission efficiency (Heinritzi et al., 2016) as well as towards its detection efficiency regarding the sulfuric acid concentration (Kürten et al., 2012).

### 2.1 Setup and Measurement Procedure

Before a more detailed technical description of the TD-DMA is given (Sect. 2.2), an overview of the measurement procedure shall be provided. The measurement procedure is discontinuous and is performed in two steps as follows:

The first step is the collection of particles (see Fig. 1a). The analyte aerosol passes through a charger, where a charge equilibrium is established. A core sampling system, where the analyte is isokinetically sampled from the center of a larger tube, is used to reduce losses in the inlet line (e.g. Wimmer et al., 2015). The aerosol then enters the DMA-unit of the TD-DMA. The gas phase of the analyte aerosol is removed by the pure nitrogen sheath gas, which is controlled in a non recirculating way. Charged particles are attracted towards the central electrode and a well-defined size enters the selection slit. Inside the central electrode, a platinum filament is placed. It is only exposed to pure nitrogen from the sheath gas and samples the preselected particles by electrostatic precipitation. The position of the filament inside the central electrode implements the collection of particles directly after classification and thereby minimizes losses. While the sampling takes place, the mass spectrometer is used to measure the gas phase chemical composition of the analyte aerosol through a separate



sampling line. It is also possible to collect the whole size distribution by turning off the DMA's sheath flow (see also Sect. 4 and 6).

Once enough particle mass is collected on the filament, the second step begins (see Fig. 1b). The inlet line of the mass spectrometer is flushed with ultrapure nitrogen. Another outlet in the inlet line works as a virtual valve that flushes out the excess nitrogen added to the sampling line while maintaining a flow of air into the sampling line. This prevents contamination of the reservoir where the sample gas is taken from and also keeps the sample gas flow rate constant, which is important for chamber or flowtube experiments. The filament is then moved into the mass spectrometer's inlet line. An electrical current is sent through the filament, which is thereby heated. The sampled particles evaporate and the vapor is analyzed by the mass spectrometer.

## 2.2 Instrument details

The main part of the TD-DMA system is the in-house developed DMA unit, where the particle size selection and collection takes place. The charger is a soft X-ray diffusion charger (TSI 3088; Tigges et al., 2015). A linear feedthrough (MDC vacuum, 152.4 mm maximum travel distance) and a stepper motor are used to translate a ceramic rod with the collection filament from its collection position to its evaporation position and back. An in-house built electronic control unit allows the precise control of all parameters, i.e. flow rates, valve positions, high voltages and the heating current for the filament. In the collection mode, the unit supplies the DMA central electrode and filament with the required positive high voltages via safe high voltage (SHV) connectors and cables. In the evaporation mode, the filament is operated with a defined heating current which can be ramped as a function of time to increase the evaporation temperature stepwise (up to about 600°C). The mass flow controllers and magnetic valves enable software controlled adjustment of the flows for the aerosol, sheath gas, carrier gas and virtual valve. Further magnetic valves are used to completely shut-off or redirect flows when necessary, e.g. during the evaporation. Two automated ball valves (Grotec OSE-M) allow for separation of the TD-DMA from the gas phase measurement during sampling mode as well as closing the aerosol inlet during evaporation mode (Fig. 1). The stepper motor moving the linear feedthrough and filament is also controlled by the electronic unit. PC control of the electronics is realized by means of USB data acquisition boards (Meilhaus 1608 and 3103) and a LabVIEW software program.

The DMA unit (see Fig. 2) is designed to collect a maximum amount of particulate mass for a defined diameter. Its characteristics are determined by the dimensions as well as the sheath- and aerosol-flow rates. High transmission efficiency is a primary goal. Also, increasing the aerosol flow rate leads to an increase in collected mass for constant transmission efficiency. Since the collected mass per time is related to the integral of the transmission efficiency as a function of size, there is always a trade off between the size resolution (high resolution means less total particles are transmitted) and the collected mass. We have therefore chosen an aerosol flow rate of 3 standard liters per minute (slm) and a sheath flow rate of 5 slm. The DMA has a cylindrical geometry (Reischl et al., 1997, Chen et al., 1998) with a positive high voltage on the central electrode (1) and ground potential on the outer electrode. The DMA's inner radius is 15 mm, the outer radius is 20 mm and the classification length is 15 mm. The collection of particles on the filament takes place inside the central electrode



(2). In this way, losses are minimized as the transfer lines are very short and the particles do not need to overcome an additional voltage step (from high voltage to ground potential) because they do not need to leave the DMA (Franchin et al., 2016). In order to position the filament, the central electrode has a through-hole in its center through which a ceramic rod is inserted. The rod carries two thick copper wires with a short piece of platinum wire spot-welded to the end of both copper

wires. This design allows depositing sufficient electric energy in the thin platinum filament when a current is applied. Electronic relays suitable for high voltage operation are used to disconnect the wires from the current source during collection mode; in this mode a third relay is used to apply a positive high voltage to the wires. While the opening in the first part (towards the linear feedthrough) of the central electrode is just large enough to center the ceramic rod while allowing for its axial movement, the through-hole in the second part (towards the DMA exit) is larger to allow moving the filament

smoothly towards the sampling line of the mass spectrometer. The outlet is closed by a ball valve during collection mode and is only opened when the filament is positioned in the CI-APi-TOF sampling line. In this mode, all other valves connected to the DMA are closed to avoid gas leaking from the DMA into the sampling line or vice versa. The shape of the collection area is an attempt to keep potential disturbances in the air flow small, while maximizing the distance between the filament and electrode to avoid flashovers and leakage currents due to the electric field. However, leakage currents could not be fully

avoided (see Sect. 3.2).

The DMA sheath flow is distributed in azimuthal direction by introducing a pressure drop at the inlet and outlet using a circular cavity and a total of 16 small drilled holes (3, 4). To achieve a laminar sheath flow, which is crucial for accurate size selection, an evenly webbed tissue (5) is placed 20 mm upstream of the selection area (6). The aerosol flow is introduced into the selection area through a narrow slit (7) with a design as recommended by Chen et al. (1998). The aerosol outlet has

to be at the side (8) and cannot be on the central axis as this is used for moving the filament to its evaporation position. In order to achieve an evenly distributed sample and outlet flow over the whole cross section of the inner part of the central electrode, a small slit introducing a pressure drop and two outlet connectors (separated by 180°) are used. The losses in this slit do not matter as the particles are sampled upstream. However, the characterization of the DMA-unit (see Sect. 3.1) is performed at the side outlets to ensure the flow conditions are the same as for the actual measurements. Therefore, note that

the actual transmission efficiencies at the filament position (i.e., the fraction of particles that reach the filament) are likely higher than determined by the characterization measurements.

A positive high voltage is applied to the central electrode by two SHV connectors since this electrode is split into two parts (10). Small spacers (11) allow changing the width of the aerosol inlet, the selection slit and the outlet, respectively. These spacers proved to be helpful for finding suitable distances for an appropriate transmission efficiency, resolution and

sizing accuracy when first testing the DMA design. The material of the DMA is stainless steel for the conductive parts (blue) and PEEK (poly-ether-ether-ketone) for the insulating parts (yellow), as these materials are considered to be chemically inert to the substances present in the aerosol sample. The full instrument measures 12 cm x 12 cm x 95 cm and weighs 13 kg.



## 3 Characterization

In this section, the general performance of the TD-DMA is described based on characterization experiments in the laboratory. These measurements are used to derive the sizing accuracy, the DMA's resolution as well as its transmission and collection efficiency. Furthermore, application of defined amounts of sulfuric acid on the filament yields information on the
detection limit of the TD-DMA in combination with the CI-APi-TOF and on the linearity between signal and collected mass.

### 3.1 DMA-Unit

The DMA unit is characterized in a tandem DMA setup (Fig. 3). Ammonium nitrate particles are produced in a nebulizer followed by a diffusion dryer. The particles are brought into charge equilibrium by using a soft X-ray charger (TSI model 3088, Tigges et al., 2015). A first DMA (Grimm Aerosol Technik GmbH model 55-100, Jiang et al., 2011) then selects
particles of a defined electrical mobility and thus corresponding size (if all particles are singly charged, which is a valid assumption for sub 30 nm particles). Note that the aerosol coming from the DMA is not strictly monodisperse; instead the DMA provides a Gaussian-shaped size distribution. This quasi-monodisperse aerosol flow is subsequently split into two flows. One flow enters the TD-DMA, after which a CPC (TSI model 3776, Hermann et al., 2007) measures the number concentration $N_1$, while the filament is removed. The other flow is directly led into a CPC which serves as a reference and
measures the concentration $N_2$. This method is based on having identical counting efficiencies for both CPCs. Therefore, the CPCs are characterized against each other for each diameter using the same setup but without the TD-DMA, and a corresponding correction factor is applied. The ratio of the two measured concentrations yields the TD-DMA's transmission efficiency τ:

$$\tau(d_P) = \frac{N_1}{N_2}.$$

(1)

At a fixed diameter selected for the first DMA, the TD-DMA voltage is varied. This method is repeated for six particle
diameters $d_P$, ranging from 7.5 to 30 nm. These measurements yield the transmission efficiency curves (Fig. 4). The error bars for the transmission efficiency are calculated using the statistical error of all individual measurements of the CPCs as well as assuming systematic errors of 10 % as suggested by the manufacturer (TSI Incorporated, 2007). The errors of the voltage set by the TD-DMA control box originate from the high voltage module and are assumed to be 5 %. As the selection slit of the TD-DMA has a non infinitesimal width, the resulting distribution is not singular but triangular to a first
approximation. Particle diffusion distorts it further and results in a Gaussian shaped curve (Stolzenburg and McMurry, 2008). The experimental data points are fit by the Gaussian function

$$\tau(U) = \tau_{max} \cdot \exp\left(-\left(\frac{U - U_{center}}{HWHM}\right)^2\right),$$

(2)



where $\tau_{\mathrm{max}}$ is the maximum transmission efficiency for each diameter, HWHM is the half width at the half maximum of the curve, and $U_{center}$ is the voltage applied to the central electrode to select the given particle size. In Fig. 5, upper panel, the fitted voltage values from the measurement are displayed as a function of the mobility diameter. The theoretical curve is calculated from the dimensions of the DMA according to Stolzenburg and McMurry (2008) and Wang and Flagan (1990).

The curve fits best when corrected by a factor of 0.81. The resaon for this is unclear, but the factor is constant for all measurements. The maximum transmission efficiency $\tau_{\mathrm{max}}$ for each diameter is shown in Fig. 5, middle panel. For 15 nm particles, it is slightly above 50 %. These values are used to estimate the collected mass when sampling.

The resolution $\mathcal{R}$ is defined as the ratio of the electrical mobility $Z$ to the width of the transmission curve at half maximum with respect to electrical mobility (Zhang and Flagan, 1996; Flagan, 2008):

$$\mathcal{R} = \frac{Z}{2 \cdot HWHM_Z}. \tag{3}$$

Deriving the electrical mobility from the diameter (Stolzenburg and McMurry, 2008; Hinds, 1999),

$$Z(d_P) = \frac{q \cdot C_C(d_P)}{3\pi\eta \cdot d_P} , \tag{4}$$

the abscissa can be transformed into mobility values, using $q$ the particle charge (for singly charged particles, $q$ equals one elementary charge), $C_C$ the Cunningham slip correction factor and $\eta$ the viscosity of the fluid. The resolution can then be derived using the fitted HWHM of the Gaussian curves. The resolution versus the diameter is displayed in Fig. 5, lower panel, which shows a maximum value of 1.19.

With a resolution of 1.19 and a transmission efficiency above 50 % at 15 nm particle size, the performance of the TD-DMA is suitable for atmospheric field studies and aerosol chamber investigations.

## 3.2 Collection Efficiency

The same tandem DMA setup as in 3.1 is used to characterize also the collection efficiency of the filament as a function of the particle diameter. For these measurements, the first DMA and the TD-DMA are set to select the same particle size. The

filament voltage is increased to the maximum possible value without influencing the voltage on the central electrode, which is also monitored continuously. Unfortunately, it is technically not possible to set the filament voltage high enough to collect 100 % of the particles for all tested particle sizes. The reason for this is most likely a leak current along the surface of the ceramic rod and thus a transfer of charge from filament to central electrode. Therefore, the filament voltage is set as high as possible with some margin to avoid an influence on the DMA high voltage.

The ratio of the two measured concentrations corrected with the DMA transmission efficiency equals the fraction of particles that are not collected on the filament. The collection efficiency is therefore defined as



$$\eta(d_P) = 1 - \frac{N_1}{N_2} \cdot \frac{1}{\tau(d_P)}, \tag{5}$$

with $\tau(d_P)$ being the transmission efficiency from Sect. 3.1. The results are displayed in Fig. 6. The efficiency at which the selected particles are collected on the filament decreases with increasing size. Since the flow velocity and the distance between electrode and filament remain constant during the sampling, the collection efficiency depends only on the particle electrical mobility for a given potential difference between filament and central electrode. However, the collection efficiency is close to unity for all sizes up to 15 nm and above 50 % for sizes up to 30 nm, which can be regarded as the current upper size limit of the TD-DMA. Therefore, it can be claimed that the collection efficiency is sufficiently high for the size range up to 30nm, and the results can be corrected with the experimentally determined values (see Sect. 5.1, Eq. 11).

**3.3 Filament Temperature**

In order to evaporate the sample in front of the mass spectrometer, the filament is heated. This is achieved by sending an electric current through the wire. Due to the rather high resistance of the filament, power is released and the filament temperature increases. Also, the platinum changes its resistance ($R_{\text{Filament}}$) due to the heating. This feature can be used to estimate the filament temperature in a similar way as in platinum thermometers such as Pt100. Here, it is not crucial to know the temperature with high precision, nevertheless it can be beneficial to determine it, e.g. in order to gain information about the volatilities of substances in an aerosol sample.

The control unit of the TD-DMA monitors the voltage $U$ and current $I$ across the filament. However, also the circuit in the electronic control unit as well as the wires connecting the filament add to the total resistance:

$$\frac{U}{I} = R_{\text{Filament}}(T) + R_{\text{System}}. \tag{6}$$

The filament resistance at room temperature (20 °C) is 0.4 Ω. Using this number together with the temperature dependent resistance of platinum,

$$R_{Filament}(T) = R(0\,°C) \cdot (1 + aT + bT^2), \tag{7}$$

with $a = 3.9083 \cdot 10^{-3}$ ΩK$^{-1}$ and $b = -5.775 \cdot 10^{-7}$ ΩK$^{-2}$ (International Electrotechnical Commision, 2008), $R(0\,°C)$ can be derived.

Then, by applying very small currents that hardly warm the filament and measuring the magnetic field resulting thereof with a current clamp, one can estimate the resistance of the system as $R_{system} = (0.1 \pm 0.1)$ Ω. With this information we can derive the temperature $T$ during all times of the evaporation process by using the feedback voltage $U$ and current $I$:



$$T(U,I) = -\frac{a}{2b} - \sqrt{\left(\frac{a}{2b}\right)^2 - \frac{1}{b} - \frac{R_{system}}{b \cdot R(0°C)} + \frac{U}{I \cdot R(0°C) \cdot b}} \; . \tag{8}$$

During standard operation, the filament is heated up to 350 °C for evaporating the sample and up to 600 °C for cleaning.

### 3.4 Sample Mass Calibration and Reproducibility

Sample mass calibration tests relate evaporated mass to mass spectrometer response. This method is used to test the evaporation method regarding its reproducibility and is required for the signal quantification (see Sect. 5.2) and determination of the detection limit. When installing the TD-DMA at the mass spectrometer e.g. after transport, they are also used to find the optimum filament position in front of the mass spectrometers ion source.

For the sample mass calibration experiments, a defined amount of a test substance is deposited on the filament and evaporated in front of the mass spectrometer. In this case, a solution of sulfuric acid ($H_2SO_4$) in water is used. After reaction with the nitrate primary ions, this substance produces different ion signals (peaks) with the main contribution coming from clusters of sulfuric acid with the primary ion monomer ($H_2SO_4\,NO_3^-$) and bisulfate ions ($HSO_4^-$). Sulfuric acid dimers ($H_2SO_4\,HSO_4^-$) as well as clusters of sulfuric acid with the primary ion dimer ($H_2SO_4\,HNO_3\,NO_3^-$) are also detected, but with a much lower intensity than the former, so that clusters with the primary ion trimer and higher are negligible.

However, for measuring internally or externally mixed particles, it should be noted that these consist of more than one substance; therefore the signals are spread over many m/z peaks, which will require more total particle mass to overcome the detection limit. In addition, the ionization efficiency of a compound will affect its detection limit in the same way as in gas phase measurements.

Figure 7a shows the typical shape of the signals resulting from the desorption process. It shows a sharp increase when a certain temperature is reached followed by an exponential decay reaching background levels within less than 30 seconds. This indicates that the substance desorbs from the filament within a narrow temperature range and that the temperature distribution of the filament is fairly homogeneous, otherwise the signal would be smeared out more strongly.

For signal analysis, the full spectrum is corrected with the relative transmission efficiency (Heinritzi et al., 2016), which takes into account that the detection efficiency inside the mass spectrometer is a function of ion mass. The individual signals $S_j$ are then normalized by the primary ions' count rates, in this case $NO_3^-$ ($m/z$ 62), $(HNO_3)NO_3^-$ ($m/z$ 125) and $(HNO_3)_2NO_3^-$ ($m/z$ 188). Subsequently, the signal is integrated over the evaporation time in order to obtain the total signal resulting of a substance $i$ from the evaporation. Finally, the signals of all relevant peaks $j$ are summed up.





$$\tilde{S}_i = \sum_{\substack{contributing \\ peaks\ j}} \int_{t=t_{start}}^{t=t_{end}} \ln\left(1 + \frac{S_j(t)}{S_{NO_3^-}(t) + S_{HNO_3NO_3^-}(t) + S_{(HNO_3)_2NO_3^-}(t)}\right) dt \qquad (9)$$

These time integrated signals are displayed in Fig. 7b and c. All data points are corrected with blank measurements obtained by evaporating only the solvent (water), without the test substance. Reproducible results are obtained for a sample mass as small as 0.01 nanograms. Additionally, the filament is heated without any sample or solvent on it to obtain a zero

measurement (see also Sect. 4.1, heating background). This zero measurement plus three times its standard deviation defines the lower detection limit. The zero measurement and the measurement points below 0.03 ng – although showing a good reproducibility – are systematically lower than the signals when larger masses are applied, which is not yet understood.

Starting from a deposited mass of 0.03 ng, the data indicates very good linearity, which is an important feature of the TD-DMA combined with a CI-APi-TOF. Furthermore, we can now relate the mass spectrometer signal to a sample mass.

For the component used here, the conversion factor $k$ is 42.95 ng (ncps·s)$^{-1}$. For other substances, their individual ionization efficiencies relative to that of the test substance need to be taken into account (relative ionization efficiency $RIE_i$, see also Sect. 5.2) and a mass spectrometer signal $\tilde{S}_i$ relates to a mass of

$$m_i = k \cdot RIE_i \cdot \tilde{S}_i. \qquad (10)$$

Concluding, the reproducibility and linearity of the evaporation process can be clearly stated.

## 4 Measurement Procedure

For measuring a nucleation event, the following procedure is desirable: At the beginning of the event, the integral measurement mode is used to analyze the freshly nucleated, smallest particles. When the mode diameter of the particle size distribution reaches around 10 nm, several size resolved measurements are performed. The sampling time is adjusted so that enough mass is collected for given particle size and concentration in the aerosol sample. Due to the minute mass contained in the nano particles, such a procedure is, however, not always possible, especially for short or weak nucleation events. In the

future, we therefore aim at improving the sensitivity as suggested in Sect. 6. For now, in case of short nucleation events or those with low concentrations, one can choose one size and sample for the whole event or use the integral mode. In either case, different kinds of background measurements are important and therefore the different steps during a full measurement cycle are described in the following.

### 4.1 Background Measurements and Signal Correction

To verify measurements and to distinguish the signal arising from particles from possible other sources, background measurements need to be taken regularly.



(1) Mass Spectrometer Background. To determine this background contribution, the nitrogen carrier gas (the flow applied to the sampling line of the mass spectrometer during the evaporation of particles) is applied to the mass spectrometers sampling line without positioning the filament inside it (Fig. 1a). This measurement accounts for the instrumental backgrounds of the CI-APi-TOF (ion source and detector) and is taken shortly before and after every evaporation. It is also used to correct the gas phase measurements of the CI-APi-TOF.

(2) Heating Background. The filament is placed at its evaporation position and heated, but without an aerosol sample. As the evaporation takes place in the inlet line that is also used for the gas phase measurements, the walls contain adsorbed material, which could re-enter into the gas phase due to the heating. The core sampling system, in which the filament is placed (see Fig. 2b), prevents most of the gas flow directly in contact with the walls from entering the ion source of the mass spectrometer. However, the inlet line downstream of the core sampling probe has been in contact with the gas phase. In the chemical settings tested, it has been observed that for filament temperatures lower than 400 °C, the signal from this process is negligible. This is the temperature range where most of the particle phase sample desorbs. For higher temperatures, the background signals increase. When observing individual ion signals as a function of time during heating, the background contribution from the sampling line walls also shows a different shape compared to the particle phase signal (Fig. 8). The signals from particles on the filament show distinct spikes with a fast increase and a rapid exponential decay, while the wall background results in slowly rising signals, which do not show a pronounced decay.

(3) Gas Phase Adsorption Background. The gas phase adsorption on the filament is corrected for by placing the filament in the sampling position inside the TD-DMA and performing the measurement in the same way as regular sampling, the only difference being that the filament is set to the same potential as the central electrode; thus, no particles are sampled. When the nitrogen sheath flow is used, as in the size resolved measurements, the filament is only exposed to nitrogen and there is hardly any relevant gas phase adsorption. However, for the integral mode, the sheath flow is turned off and the filament is thus exposed to the gas phase of the aerosol sample. When heating the filament, the signals show the same shape as for the particle measurements, as they are originating from the filament itself. This background measurement is taken for every major change in gas phase chemistry. For most of our tested systems, this background was comparably low and mostly negligible.

## 4.2 Example Measurement

The instrument was tested during the CLOUD10T campaign at the Cosmics Leaving Outdoor Droplets (CLOUD) chamber at the European Organization for Nuclear Research (CERN). CLOUD is a 26 m$^3$ stainless steel chamber used for investigating nucleation and growth of aerosol particles under atmospherically relevant and precisely controlled conditions (Kirkby et al., 2011). Figure 8 shows a measurement of size selected 15 nm particles generated from 119 ppt$_v$ of dimethylamine, $4.3 \cdot 10^6$ cm$^{-3}$ sulfuric acid, at a temperature of 4.8 °C and relative humidity of 40.2 %. The total particle number concentration (> 2.5 nm) is approximately $1.7 \cdot 10^5$ cm$^{-3}$ and the TD-DMA collected its analyte particles for a duration of 60 minutes. While the filament is gradually heated, substances desorb and are subsequently ionized by the





negative nitrate primary ions and detected by the mass spectrometer. The example substances shown here desorb at different temperatures. The signal shape for a given ion signal as a function of time is the same as for the sample mass calibration (Sect. 3.4, Fig. 7); a sharp increase is followed by an exponential decrease, as the material on the filament is depleted. At filament temperatures beyond 350 to 400 °C, signals of different shapes appear (in this case e.g. m/z 147 and m/z 163).

These signals are also found in the heating background measurements and do not originate from the collected analyte particles. Instead, they are caused by desorption of substances from the inlet line due to the increased temperature of the carrier gas. The fact that the inlet line is a much larger reservoir than the filament explains why the shape of these background signals is different and why they do not decrease within a few seconds. The temperature range where valid measurements are taken is thus up to ~350 °C. Nevertheless, the filament is heated up to 600 °C in order to ensure that all

particulate material evaporates and memory effects are avoided.

Some substances seem to desorb at two different temperatures. This can happen for two reasons: 1) The filament might be heated unevenly, but this would also result in less pronounced peaks instead of two sharp peaks. 2) Different chemical species that do not possess the same evaporation temperature, can result in the same product ions. In the example shown here, it is not exactly clear whether sulfuric acid belongs to dimethylaminium-bisulfate or dimethylaminium-sulfate,

and what influence potential contaminants such as ammonia can have on the evaporation temperature (Lawler et al., 2016). 3) The signal appearing first originates from desorption of the molecule with this given mass to charge ratio, whereas the second appearance could be due to the fragmentation of a larger molecule. Some amount of the larger molecules seem to fragment and these fragments are then detected at higher temperatures than expected for a desorption (see also Sect. 6). In the chemical system of dimethylamine and sulfuric acid, one can also observe the selectivity of the ionization by nitrate ions.

Sulfuric acid, for example, has a lower detection limit of $5 \cdot 10^4$ cm$^{-3}$ (~0.00185 ppt$_v$) (Kirkby et al., 2016), whereas for dimethylamine the lower detection limit of 1.7 ppt$_v$ (Simon et al., 2016) is almost a factor of $10^3$ higher. Therefore, the signals related to sulfuric acid are very strong whereas DMA related signals mostly do not reach the detection limit.

## 5. Signal Quantification

### 5.1 Collected Particle Mass During a Measurement

The collected mass on the filament is calculated in order to quantify the measured signals. Additionally, it is estimated before the measurement from the present particle size distribution and the selected particle diameter so that the collected particle mass exceeds the detection limit.

As a DMA cannot provide an absolute monodisperse aerosol (see Sect. 3.1), the incoming number size distribution $dn/dd_p$ of the sample in combination with the transmission efficiency $\tau(d_p)$, collection efficiency $\eta(d_p)$ and charge fraction

$x(d_p)$ is needed to calculate the collected mass precisely. The transmission and collection efficiency are retrieved from the TD-DMA characterization (Sect. 3, Fig. 4, Fig. 5, and Fig. 6)  and the incoming size distribution can be provided by an



SMPS or other instrument measuring the number size distribution of the analyte aerosol in the applicable size range. Thus, the collected mass on the filament can be calculated according to:

$$m_{coll}(d_p') = Q_{aerosol} \cdot \rho \cdot \frac{\pi}{6} \cdot \int\limits_{t=t_0}^{t_{coll}} \int\limits_{d_p=0}^{d_{p,max}} d_p^3 \cdot \frac{dn(d_p,t)}{dd_p} \cdot x(d_p) \cdot \eta(d_p) \cdot \tau_{d_p'}(d_p) \cdot dd_p \cdot dt, \qquad (11)$$

with $Q_{aerosol}$ the aerosol flow rate through the TD-DMA, $\rho$ the particle density and $d_p'$ the selected diameter. The amount of mass collected per time therefore depends both on the selected diameter and on the size distribution of the aerosol analyte. In addition, as for any DMA, the effective diameter collected varies with the size distribution of the background aerosol. To get an impression of the amount of mass that can be collected under certain conditions, we look at the aerosol size distribution of the example event described in Sect. 4.2 (see Fig. 9). The lower panel displays the number size distribution and median diameter (black line), as well as the time period of collection (pink line). The upper panel shows the particle mass collected by the TD-DMA per unit time when the instrument is set to collect a certain diameter; it ranges between $10^{-5}$ ng s$^{-1}$ and $10^{-3}$ ng s$^{-1}$. Integration over time yields the total mass collected as described by equation (11).

The minimum mass concentration in the analyte aerosol where measurements with the TD-DMA are possible, depends not only on size distribution and selected size but also on sampling time and analyte aerosol composition. For example, the lower limit for sulfuric acid particles with a sampling time of 2 hours is 27 pg/m$^3$. Measuring in size resolved mode from monodisperse 15 nm particles, 811 pg/m$^3$ are needed. Assuming the number size distribution from the example nucleation event (Fig. 8) and measuring all sizes, the mass concentration should be higher than 385 pg/m$^3$. The TD-DMA's sensitivity is thus comparable to other instruments operating in this size range.

### 5.2 Mass Fraction of Individual Substances from Time Integrated Signals

The fraction of a substance in the aerosol sample shall be determined. Like this, the relative contribution of an individual substance or a group of substances (e.g. all organic compounds) to nucleation can be investigated, e.g. for the same size under different conditions or with increasing size under constant conditions.

The calculation requires the collected mass from Sect. 5.1 and the conversion factor from the sample mass calibration in Sect. 3.4. The fraction $f_i$ of an individual substance in the aerosol sample is then

$$f_i = \frac{m_i}{m_{coll}} . \qquad (12)$$

Note that the collected mass calculated from the SMPS data should be used instead of the sum of all identified components, $\sum_i m_i$. Using the latter would be based on the assumption that all compounds of the aerosol could be charged, detected, identified and quantified. As we know from gas phase measurements, this is usually not the case for the chemically complex atmospheric aerosols. Chemical ionization, especially the negative nitrate ionization used here, is a selective technique. A substance needs to react with the primary ions to be charged and detected in the mass spectrometer. This can either be the





transfer of a positively charged hydrogen atom from the substance to the primary ion, or clustering with the primary ion (Hyttinen et al., 2017). For the same reason, the signal of each substance also needs to be corrected with its ionization efficiency (see Sect. 3.4) and this property is not known for all substances. That circumstance can also be bypassed by comparing the contribution of compounds under different conditions rather than providing an absolute concentration.

## 6 Discussion

The detection limit and sensitivity are a limit to the parameters particle size, number concentration of the analyte aerosol, ionization efficiency, complexity of the spectrum and time resolution. A major challenge is the fact that small particles have an extremely low mass. For example, a particle of half the diameter contains a factor of eight less mass. In addition, charging probability decreases with size. At 15 nm, the fraction of singly negatively charged particles is less than 0.07 (Fuchs, 1963; Tigges et al., 2015) for the soft x-ray charger used here. For an aerosol of multi component composition, it applies that the more particle mass is collected, the more substances overcome the lower detection limit and the more detailed the observed mass spectrum will be. The TD-DMA is thus designed for high transmission at a coarse resolution. In this way, a size selection is possible but still a sufficient amount of mass is collected. In case of very low particle concentrations or when a high time resolution is needed, it is also possible to sample all particle sizes in an integral mode by turning off the sheath flow. The low mass concentration of the particulate matter is also the reason why a discontinuous system with consecutive collection and evaporation is chosen. On the other hand, particles of larger diameters bear a higher probability for being multiply charged, so that particles of two times the diameter and thus eight times the mass contaminate the sample. This can be corrected for when counting, but not in the chemical analysis. Within the size range of 10 to 30 nm that the TD-DMA is specialized for, multiple charging does not play a significant role. The soft x-ray charger was chosen for that reason and due to the fact that it does not change the chemistry much, compared to e.g. a corona charger. Nevertheless, it would be beneficial to find a charger which offers a higher charging probability for small particles while still fulfilling the above criteria.

For optimizing the collection process, it would be beneficial to enhance the collection efficiency (see Sect. 3.2) also for larger sizes. Applying a larger collection voltage to the filament increases the collection efficiency, which is currently limited by leak currents occurring between filament and central electrode when the potential difference is too high. It is also a possibility to prepare the DMA for a higher flow of the analyte aerosol in order to collect more mass.

The exact position of the filament in combination with the flow profiles at the place of evaporation is crucial and optimization of these features have a high impact on the sensitivity. Also, the high temperature gradient when heating the filament releases the analyte in a very compact way, which benefits the lower detection limit. This method of direct heating ensures a homogeneous and precisely controlled temperature for the desorption of the substances. Currently we have a discrepancy between the TD-DMA's sample mass calibration and the gas phase calibration of sulfuric acid (Kürten et al., 2012). This can be explained by losses in the transition from filament to mass spectrometer due to condensation on the mass




spectrometer's inlet line. There have also been tests to evaporate the sample directly inside the ion source, but for the presently used ion source (Kürten et al., 2011) this was not expedient. Nevertheless, these losses are unsatisfactory and possibilities for further improvements are being investigated.

As discussed in Sect. 4.2, the filament material might be problematic in terms of influencing the analyte. Heating, especially from metallic surfaces, might lead to fragmentation or other chemical modification of the original molecules. We tested different filaments and saw that not only the material but also the way the wire is manufactured strongly influences this process and therefore the suitability of the filament. Also, it was found that a careful heating procedure seems to reduce these effects. This was investigated by using the sample mass calibration method (see Sect. 3.4) with different substances and by careful examination of the spectra from nucleation events. Some fraction of the larger molecules (for example of the HOMs) seem to fragment and these fragments are then detected at higher temperatures than molecules that have the same elemental composition. A detailed analysis of this issue is subject of future studies. A complete degeneration of substances is hardly observed for the relevant temperature range, only very small peaks that might originate from such reactions are present in the spectra at higher temperatures. Nevertheless, it can not be excluded that some material breaks up into substances that are not ionized by the negative nitrate CI-APi-TOF and are thus not visible in the spectra. By coating the filament with a thin layer of a chemically inert material, this could be reduced.

The ionization with negative nitrate primary ions is a selective technique. To target more substances like e.g. Amines and Ammonia, different primary ions can be chosen (Lawler et al., 2016). To distinguish the aerosol sample from potential gas phase adsorption, a sheath flow of pure nitrogen surrounds the filament in the size selective mode. In the integral mode, the filament is also exposed to the gas phase of the analyte and a gas phase background measurement is advised (see Sect. 4.1).

Up to now, the TD-DMA has been used in chamber experiments with timed nucleation events, but it is also intended for field use in the future. Here, the size selective mode is especially useful because the larger background particles would otherwise dominate the mass concentration. Also, background measurements as suggested in Sect. 4.1(3) are not straightforward when chemical conditions change over time. To look further into the chemistry of aerosol growth, it will be interesting to perform measurements at a flow tube, where a steady state production of the analyte aerosol is possible. When not being limited by the time resolution or small number concentrations, one can analyze particles starting at small sizes with a high size resolution – which is variable in this method – and thus gain a very detailed picture of the growth process.

## 7 Summary

The TD-DMA is suitable for measuring nucleation and Aitken mode particles and is successfully analyzing aerosol mass concentrations in the pg/m$^3$ range. The DMA unit is optimized for a high transmission at a coarse resolution to collect a sufficient amount of mass. The characterization of the DMA unit and the evaporation of a test substance show to be reproducible. The instruments ability to measure freshly nucleated particles was proven in chamber experiments.




The advantages of the instrument are as follows: a) The TD-DMA allows for size selection, therefore the chemical composition of different particle sizes can be compared in order to determine the relative importance of different vapors for particle growth as a function of size. b) An integral, non size selective mode of operation is possible in order to maximize the mass of collected particles. c) The TD-DMA is a modular and compact unit that can in principle be used with different mass

spectrometers or other gas analyzers. d) During the particle collection process, the gas phase can be analyzed. For this reason, the same mass spectrometer can be used for particle as well as gas phase measurements and both phases can be directly compared. With this instrument, the processes affecting particle composition and growth can be investigated in detail.

**Acknowledgements**

We thank the workshop team of our institute for manufacturing the components and for technical support. We appreciate the useful discussion with Jim Smith from University of California, Irvine, USA, and Alexander Vogel, Institute for Atmospheric and Environmental Science, Frankfurt, Germany. We also thank Jasmin Tröstl and Urs Baltensperger from Paul Scherrer Institute, Villigen, Switzerland, for providing a nSMPS and the whole CLOUD team for support during the CLOUD10T measurement campaign.

This research was supported by the German Federal Ministry of Education and Research (project "CLOUD-12" 01LK1222A and "CLOUD-16" 01LK1601A) as well as the EC Seventh Framework Programme (MC-ITN "CLOUD-TRAIN" 316662).

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





| Method | Reference | Size selective | Continuous or Discontinuous | Evaporation method | Phase(s) measured |
|---|---|---|---|---|---|
| VACA | Curtius et al., 1998 | no | continuous | thermal desorption | gas & particle * |
| TDCIMS | Voisin et al., 2003 | yes | discontinuous | thermal desorption | particle |
| NAMS | Wang et al., 2006 | yes | continuous | laser ablation | particle |
| Aerosol MS | Laitinen et al., 2009 | yes | discontinuous | laser ablation | particle |
| Aerosol Inlet | Phares and Collier, 2010 | yes | discontinuous | thermal desorption | particle |
| CAChUP | Gonser and Held, 2013 | yes | discontinuous | thermal desorption | particle |
| FIGAERO | Lopez-Hilfiker et al., 2014 | no | discontinuous | thermal desorption | gas & particle |
| TD-DMA | *this study* | yes | discontinuous | thermal desorption | gas & particle |

*not separated

**Table 1:** Instruments capable of chemical analysis of sub 30 nm particles.





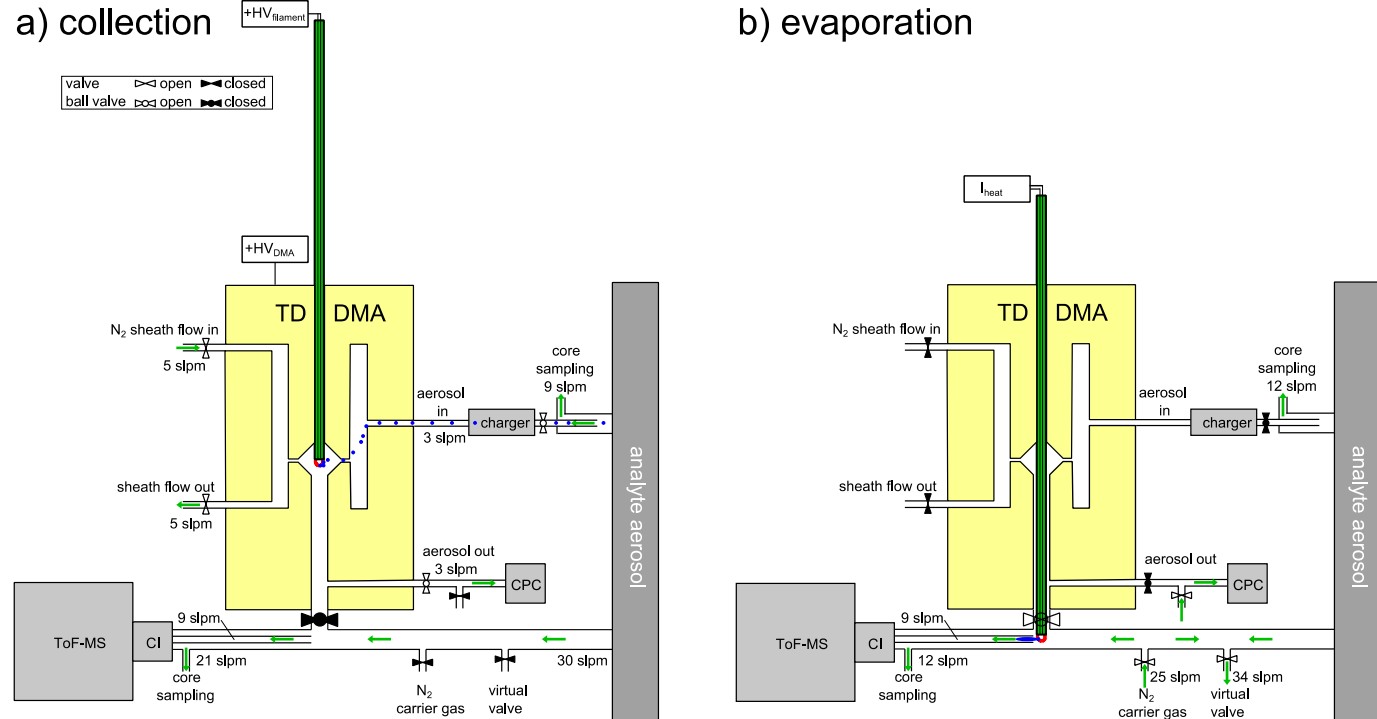

**Figure 1:** Measurement setup of the TD-DMA with step a) collection, and b) evaporation. Filled valves correspond to closed, unfilled ones to open. In step a, particles (blue) from the aerosol sample are charged, size selected and collected on a filament (red) inside the DMA-unit (yellow). In step b, the filament is moved in front of a mass spectrometer and the collected material is evaporated and analyzed.





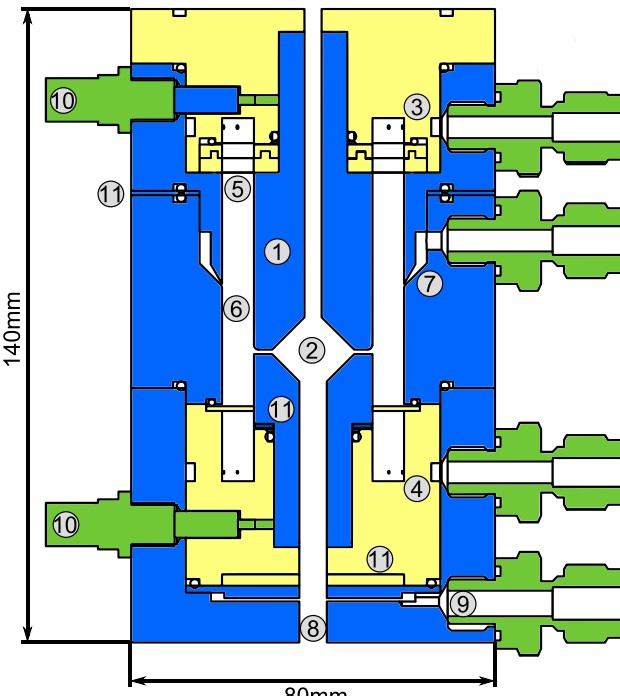

**Figure 2:** Schematic quarter section of the DMA unit of the TD-DMA with 1) central electrode, 2) collection area, 3) sheath flow inlet, 4) sheath flow outlet, 5) tissue for laminarising the sheath low, 6) selection area, 7) aerosol inlet, 8) outlet for the filament, 9) aerosol outlet, 10) SHV supplies and 11) spacer plates. Conductive stainless steal parts are shown in blue, isolating PEEK parts in yellow.



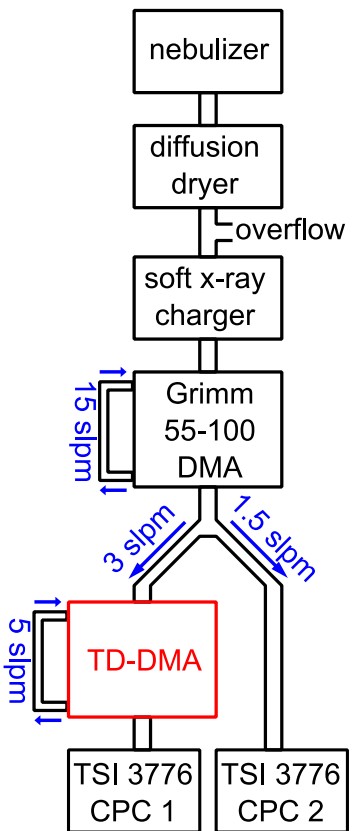

**Figure 3:** Tandem DMA setup for the characterization of the TD-DMA's DMA unit.





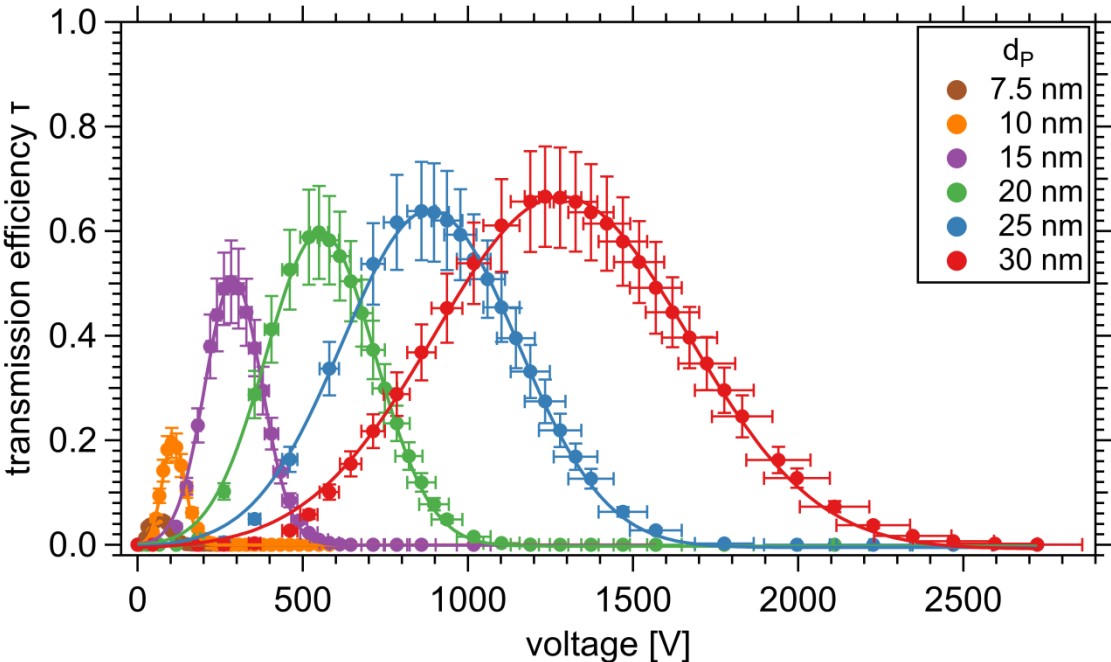

**Figure 4:** Transmission efficiency of the TD-DMA for different diameters. The DMA unit is characterized in a tandem DMA setup.




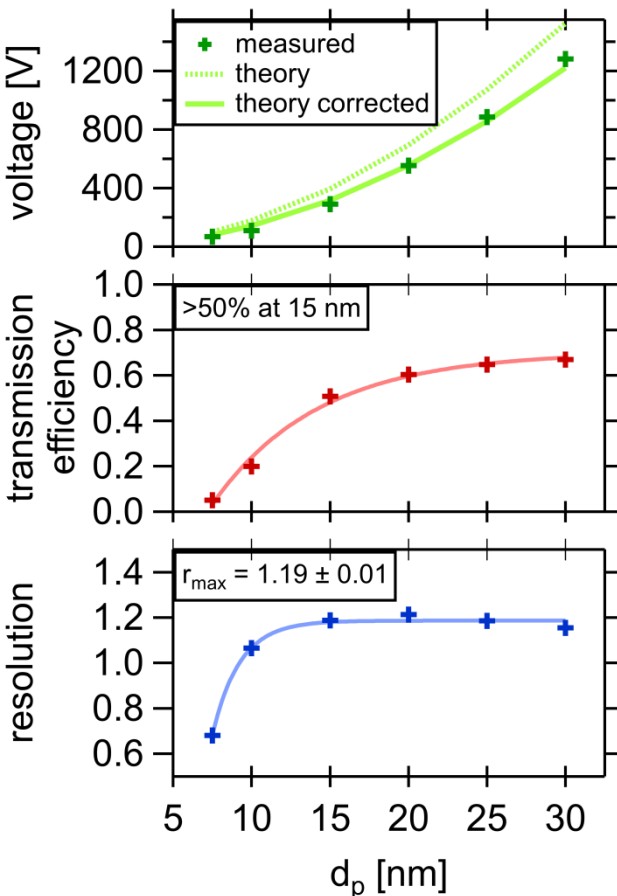

**Figure 5:** Summary of DMA performance parameters. The DMA is optimized for a coarse resolution and high transmission in order to collect a maximum amount of particle mass while still allowing size selection.





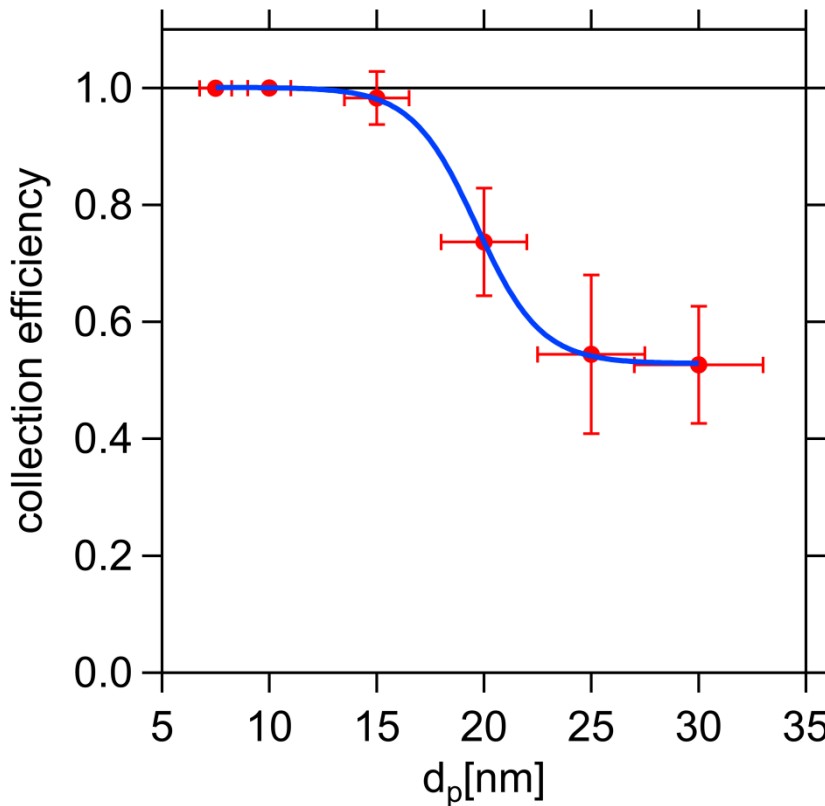

**Figure 6:** Collection efficiency depending on the particle size. Due to the decreased electrical mobility, the required collection voltage increases with particle size, which currently limits the collection efficiency at larger sizes.



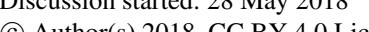

**Figure 7:** Sample mass calibration with a) typical desorption shape of signals, c) integrated signal vs. injected mass, and b) zoom on lower masses with detection limits.





**Figure 8:** Composition measurement of 15 nm particles generated from dimethylamine, sulfur dioxide and ozone in presence of UV light. a) Spectrum of particle phase compared to the background. Prominent particle phase peaks in black, primary ions in grey. b) Time series of evaporation. With increasing filament temperature, substances desorb from the filament and are depleted rapidly. At high temperatures, signals originating from the inlet line appear.





**Figure 9:** Particle mass collected by the TD-DMA per unit time when the instrument is set to collect a certain diameter (upper panel), for a given size distribution (lower panel).





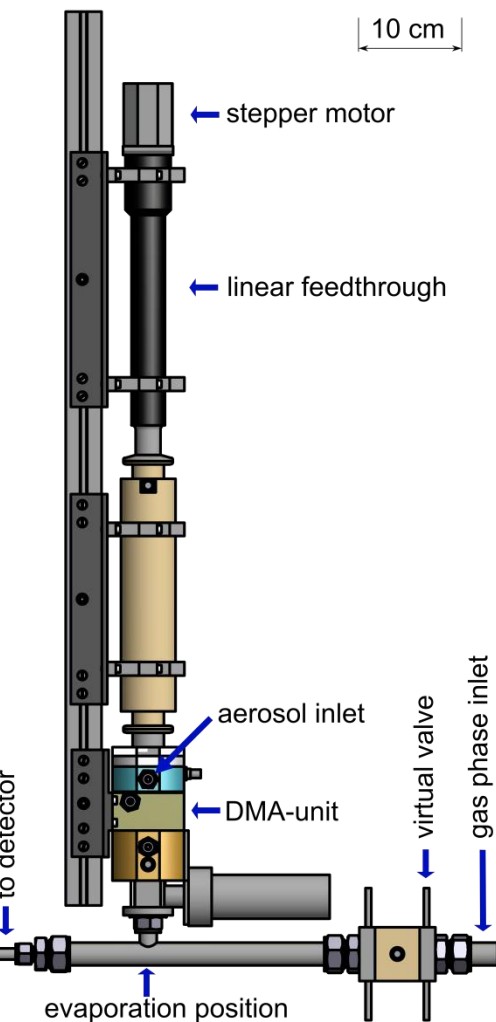

**Figure 10:** The TD-DMA coupled to a gas phase detectors inlet line.