# Peer review of "Size Resolved Online Chemical Analysis of Nano Aerosol Particles: A Thermal Desorption Differential Mobility Analyzer Coupled to a Chemical Ionization Time Of Flight Mass Spectrometer"

_Atmospheric Measurement Techniques, 2018_

## Referee Comment (RC1) · Anonymous Referee #1 · 20 Jun 2018

This manuscript (amt-2018-116) describes the development of a thermal desorption differential mobility analyzer (TD-DMA) coupled to a chemical ionization time-of-flight (CI-TOF) mass analyzer for the analysis of ambient nanoparticles down to ~10 nm diameter. Nanoparticles are charged with an X-ray source, size selected in a custom DMA, and collected onto a metal filament. After sufficient particle mass has been collected, the filament is resistively heated to desorb the collected material, which is subsequently analyzed by CI-TOF. This approach provides a size resolution of 1.19, a transmission efficiency of 50% at 15 nm, and a detection limit of about 10 pg aerosol

mass.

The manuscript represents a new approach to analyze the composition of ambient nanoparticles, which is significant because very few approaches currently exist. The manuscript is within the scope of Atmospheric Measurement Techniques and will be suitable for publication if the below comments are carefully addressed. These comments generally revolve around provide more details about the instrumental approach and its effectiveness as well as better placing this instrumental approach in the context of existing nanoparticle chemical analysis approaches.

Comments:

1. The first major comment relates to insufficient experimental details and justification. For example, on page 5, lines 28-31, the authors that there is a trade off between size resolution and collected mass that underlies their choice of aerosol flow and sheath flow rates. However, they provide no additional detail as to how they arrived at that choice: how much have the authors sacrificed in size resolution to increase collected mass? A second example relates to aerosol charging. As the authors acknowledge on page 15, lines 16-19, multiply charged aerosol could compromise their measurement as a doubly charged particle with the same mobility as a singly charged one has about 8 times more mass to it. Only a small number of multiply charged particles are required to significantly bias the composition measurement. The authors discount this possibility by simply stating "multiple charging does not play a significant role" (page 15, line 19). In their revision, the authors need to provide significant more justification for this statement as its accuracy determines whether this approach is actually sampling particles at the size they claim.

2. The second major comment relates to placing this new approach in the context of other existing approaches. The authors have attempted to do this to some extent in Table 1. However, in their revision they must provide additional comparison to existing measurements. For example, perhaps the most commonly used approach similar to

this instrument is the thermal desorption chemical ionization mass spectrometer (TD-CIMS), which also charges the aerosol, size selects using DMAs, collects it onto a filament, which is then resistively heated to desorb molecules that are then ionized and detected. As far as this reviewer can ascertain, the main differences in these two approaches are 1) this approach uses a bipolar charger whereas TDCIMS uses a unipolar charger, 2) the DMAs used in each approach may be configured differently, and 3) TD-CIMS uses water clusters whereas TD-DMA uses nitrate clusters for ionization. The impact of this work would be significantly enhanced if the authors discussed the similarities and differences between their instrument and others (like TDCIMS), providing details about how charging efficiencies, multiple charging, detection limits, time resolution, and ionization efficiency differ based on the instrumental configuration. At a minimum, in their revision the authors should include, perhaps in an additional table, key parameters describing instrument performance (e.g. detection efficiency/sensitivity, aerosol mass collection rates, etc.) for their instrument as well as available literature data for the other approaches.

3. The authors highlight as a key benefit of their instrument that they can perform gas phase measurements as well as particle phase measurements. However, virtually no further details are provided, and the instrument does not appear to have been used to investigate partitioning in the example study in the manuscript. In their revision, the authors should provide some additional details about the benefits of being able to do both measurements with their instrument, as that is a unique aspect of their instrument. Related to this point, on page 4, line 30, the authors do not indicate whether this gas sampling line has a filter to remove any aerosol that might bias the measurement. The authors should clarify this point in their revision.

4. Some of the language used in the manuscript is imprecise. One example of this is in the discussion of the sampling setup on page 6, lines 1-15. In this section, what does "electrical energy" (page 6, line 5) mean? Does it refer to heat, or voltage? The authors also describe different "parts" of the DMA (page 6, lines 7-9), but don't make

use of their labels in the figure, resulting in this section being difficult to follow. In their revision, the authors should carefully read through their manuscript and improve the precision in language.

---

## Referee Comment (RC2) · Anonymous Referee #2 · 4 Jul 2018

General Comments

This manuscript describes the design and operation of a TD-DMA-CIMS. While I would not call the technique novel (as thermal desorption-chemical ionisation mass spectrometry has been demonstrated and utilized in both laboratory and field studies) the authors have developed a new DMA enabling particle collection and heating in-situ and overall they do a good job characterising the instrument. My main concern, noted below, is that the authors do not compare their TD-DMA-CIMS implementation directly to prior TD-CIMS works and in their characterisation, they do not explain some critical assumptions regarding DMA characterisation.

Specific Comments:

1. The main issue I see is with a lack of comparison to prior TD-CIMS implementations. The authors do acknowledge the original TD-CIMS work (Voisin et al, 2003), but subsequent development and characterization efforts (Smith et al. 2004, McMurry et al. 2009) are overlooked. I think the authors need to make clearer in this study what is new/unique about their implementation of DMA-thermal desorption-chemical ionisation mass spectrometry. How does their limit of detection compare to prior implementations? I understand that the authors are able to make simultaneous particle and gas phase measurements, but I think this would also be possible with prior implementations of this technique.

2. Table 1 and lines 7-27. The table omits techniques where electrospray ionisation is utilized to generate ions. This is an important ionization technique which minimally fragments parent ions, and while it is sparingly used in aerosol science, there are a number of studies demonstrating its potential in aerosol analysis. I would recommend adding electrospray ionisation based methods to table 1 and mentioning them in the text as these methods can be applied to sub-30 nm particles. Specifically , He et al (2015) showed that nanoparticles can be collected electrostatically and then ions can be generated via electrospray, Horan et al (2017) showed that electrospray-like mass spectra can be collected for aerosol particles without the need for a distinct collection step, and SESI (secondary electrospray ionisation, sometimes called extractive electrospray ionisation), though it cannot distinguish between gas and particle phase, has been employed in several studies (Doezema et al. 2012, Gallimore and Kalberer 2013).

3. Equation (1) and the surrounding analysis. While the authors do note "Note that the aerosol coming from the DMA is not strictly monodisperse; instead the DMA provides a Gaussian-shaped size distribution," it does not appear they account for this in their analysis or explain to readers why they do not need to account for this. Equation (1) is not the true transfer function of the DMA; this would only be the transfer function if the first DMA had extremely high resolution relative to the test DMA. Looking at the sheath flow rate employed in the TD-DMA relative to the aerosol flowrate, this is probably the case and the authors' analysis is ultimately fine, but I would advise the authors to show this more clearly. Specifically, the number concentration of particles at the outlet of the first DMA is given by the equation:

$$N_2 = \int_0^\infty \frac{dn}{dd_p} \tau_G(d_p) dd_p$$

where $\tau_G(d_p)$ is the Grimm DMA transfer function/transmission function. For particles exiting the TD-DMA, the number concentration is:

$$N_1 = \int_0^\infty \frac{dn}{dd_p} \tau_G(d_p)\tau(d_p)dd_p$$

The ratio $N_1/N_2$ is hence not given by equation (1), but is:

$$\frac{N_1}{N_2} = \frac{\int_0^\infty \frac{dn}{dd_p}\tau_G(d_p)\tau(d_p)dd_p}{\int_0^\infty \frac{dn}{dd_p}\tau_G(d_p)dd_p} \neq \tau(d_p)$$

A common (reasonable) assumption is that $\frac{dn}{dd_p}$ is a constant over the region where $\tau_G(d_p)$ takes non-zero values. This leads to:

$$\frac{N_1}{N_2} = \frac{\int_0^\infty \tau_G(d_p)\tau(d_p)dd_p}{\int_0^\infty \tau_G(d_p)dd_p} \neq \tau(d_p)$$

Therefore, the method noted in the manuscript at present does not yield $\tau(d_p)$ unless $\tau_G(d_p)$ is significantly narrower (higher resolution) than $\tau(d_p)$. At an aerosol to sheath flow ratio of 3/5, I assume the assumption is reasonable, but does need to be justified or better yet, explicitly shown in the manuscript.

Doezema, L. A., T. Longin, W. Cody, V. Perraud, M. L. Dawson, M. J. Ezell, J. Greaves, K. R. Johnson & B. J. Finlayson-Pitts (2012) Analysis of secondary organic aerosols in air using extractive electrospray ionization mass spectrometry (EESI-MS). *RSC Advances,* 2**,** 2930-2938.

Gallimore, P. J. & M. Kalberer (2013) Characterizing an Extractive Electrospray Ionization (EESI) Source for the Online Mass Spectrometry Analysis of Organic Aerosols. *Environmental Science & Technology,* 47**,** 7324-7331.

He, S. Q., L. Li, H. X. Duan, A. Naqwi & C. J. Hogan (2015) Aerosol Analysis via Electrostatic Precipitation-Electrospray Ionization Mass Spectrometry. *Analytical Chemistry,* 87**,** 6752-6760.

Horan, A. J., M. J. Apsokardu & M. V. Johnston (2017) Droplet Assisted Inlet Ionization for Online Analysis of Airborne Nanoparticles. *Analytical Chemistry,* 89**,** 1059-1062.

McMurry, P. H., A. Ghimire, H.-K. Ahn, H. Sakurai, K. Moore, M. Stolzenburg & J. N. Smith (2009) Sampling Nanoparticles for Chemical Analysis by Low Resolution Electrical Mobility Classification. *Environmental Science & Technology,* 43**,** 4653-4658.

Smith, J. N., K. F. Moore, P. H. McMurry & F. L. Eisele (2004) Atmospheric Measurements of Sub-20 nm Diameter Particle Chemical Composition by Thermal Desorption Chemical Ionization Mass Spectrometry. *Aerosol Science and Technology,* 38**,** 100-110.

---

## Author Comment (AC1) · 15 Aug 2018

**Reply to Referee 1**

**"Size Resolved Online Chemical Analysis of Nano Aerosol Particles: A Thermal Desorption Differential Mobility Analyzer Coupled to a Chemical Ionization Time Of Flight Mass Spectrometer"**

We would like to thank the reviewer for the comments, which led to an improvement of our manuscript. We addressed the points carefully as described below. Each of the referee's comments is repeated in black font, while the reply is shown in blue font. Changes in the revised manuscript are written in green font.  Attached please also find the revised manuscript.

Best regards,
Andrea Wagner and Coauthors

**1** The first major comment relates to insufficient experimental details and justification. For example, on page 5, lines 28-31, the authors that there is a trade off between size resolution and collected mass that underlies their choice of aerosol flow and sheath flow rates. However, they provide no additional detail as to how they arrived at that choice: how much have the authors sacrificed in size resolution to increase collected mass? A second example relates to aerosol charging. As the authors acknowledge on page 15, lines 16-19, multiply charged aerosol could compromise their measurement as a doubly charged particle with the same mobility as a singly charged one has about 8 times more mass to it. Only a small number of multiply charged particles are required to significantly bias the composition measurement. The authors discount this possibility by simply stating "multiple charging does not play a significant role" (page 15, line 19). In their revision, the authors need to provide significant more justification for this statement as its accuracy determines whether this approach is actually sampling particles at the size they claim.

The first part of the comment refers to the choice of the flow rates for the DMA unit (sheath and aerosol flow rate). We arrived at the choice of using 5 slpm (standard liter per minute) sheath flow rate and 3 slpm aerosol flow rate as follows: First, for a given resolution, a high aerosol flow rate yields a high amount of analyte aerosol particles being selected. For the dimensions of the DMA unit, 3 slpm is the maximum aerosol flow rate that is experimentally feasible. Second, a low sheath flow rate yields a lower resolution, which results in more overall mass being collected. However, we found that if the sheath flow rate is too low compared to the aerosol flow rate, the particle selection does not work anymore. The sheath flow rate of 5 slpm was determined as the minimum for a reliable size selection for this DMA unit and with 3 slpm aerosol flow rate.

To explain this in more detail, we modified the text on page 6, lines 7-15 as follows:

"The DMA unit (see Fig. 2) is designed to collect a maximum amount of particulate mass for a defined diameter. Its characteristics are determined by the dimensions as well as the sheath and aerosol flow rates. The amount of collected particles is maximized when the transmission efficiency is high. Also, increasing the aerosol flow rate leads to an increase in collected mass for constant transmission efficiency. Hence we chose the maximum possible aerosol flow rate for the dimensions of the DMA unit, 3 standard liters per minute (slpm). As the overall number of selected particles is higher when the size resolution of the DMA is low, a low size resolution and thus low sheath flow rate is beneficial in this case. However, still the size resolution needs to be high enough to warrant size selection and at too low values of the sheath flow rate, the DMA will not work anymore. With a sheath flow of 5 slpm, as much particle mass as possible is collected while still allowing sufficient size selection. "

The second part of the comment relates to the occurrence of multiply charged particles. We now provide justification for the statement that multiple charging does not play a significant role in the TD-DMA's size range, by estimating the contribution with an example calculation. We add on page 16, lines 3-10:

"A typical particle size selected in the TD-DMA is 15 nm. Doubly charged particles of the same electrical mobility have a diameter of 21 nm (Stolzenburg and McMurry, 2008; Hinds, 1999). Considering the charging probabilities of the used soft-x-ray charger (Tigges et al., 2015), a fraction of $7.6 \cdot 10^{-2}$ of the 15 nm particles carry one charge

whereas just a fraction of $2.6 \cdot 10^{-4}$ of the 21 nm particles carry a double charge. Thus, at maximum only a fraction of 0.009 of the particle mass in the sample would originate from doubly charged larger particles. This assumes a uniform particle number size distribution. As for new particle formation events, the smallest particles have the highest number concentration, this estimate calculation gives an upper limit. In case of very large background particles in the accumulation mode size range, as for example in field use, an impactor should be used. "

**2** The second major comment relates to placing this new approach in the context of other existing approaches. The authors have attempted to do this to some extent in Table 1. However, in their revision they must provide additional comparison to existing measurements. For example, perhaps the most commonly used approach similar to this instrument is the thermal desorption chemical ionization mass spectrometer (TDCIMS), which also charges the aerosol, size selects using DMAs, collects it onto a filament, which is then resistively heated to desorb molecules that are then ionized and detected. As far as this reviewer can ascertain, the main differences in these two approaches are 1) this approach uses a bipolar charger whereas TDCIMS uses a unipolar charger, 2) the DMAs used in each approach may be configured differently, and 3) TDCIMS uses water clusters whereas TD-DMA uses nitrate clusters for ionization. The impact of this work would be significantly enhanced if the authors discussed the similarities and differences between their instrument and others (like TDCIMS), providing details about how charging efficiencies, multiple charging, detection limits, time resolution, and ionization efficiency differ based on the instrumental configuration. At a minimum, in their revision the authors should include, perhaps in an additional table, key parameters describing instrument performance (e.g. detection efficiency/sensitivity, aerosol mass collection rates, etc.) for their instrument as well as available literature data for the other approaches.

We agree that comparing existing approaches for the chemical analysis of sub 30 nm particles is very interesting and will greatly enhance the impact of this work. However this is not straight forward. Nevertheless, we have considered the reviewer's comment by adding the suggested table that provides an overview of the key parameters for the existing techniques, and by adding a distinct section to the paper discussing this comparison (section 6.2). Starting on page 17, line 17, we add:

[revised manuscript text omitted]

To link this section in the introduction, we added on page 3, line 31:

"A more detailed discussion is given in Sect. 6.2. ",
and in the quantification section 5.1, page 14, line 28:
"(see Sect. 6.2)"

As the discussion section 6 is now split up , we rename the previous discussion starting on page 15 line 16 as follows:
"6 Discussion"
"6.1 Discussion on the newly developed TD-DMA"
"6.2 Comparison of Instruments Capable of Chemical Analysis of Sub 30 nm Particles"

The work providing the sensitivity for the VACA used in the table (Arnold et al., 1998), is referred to on page 3, line 14.

**3** The authors highlight as a key benefit of their instrument that they can perform gas phase measurements as well as particle phase measurements. However, virtually no further details are provided, and the instrument does not appear to have been used to investigate partitioning in the example study in the manuscript. In their revision, the authors should provide some additional details about the benefits of being able to do both measurements with their instrument, as that is a unique aspect of their instrument. Related to this point, on page 4, line 30, the authors do not indicate whether this gas sampling line has a filter to remove any aerosol that might bias the measurement. The authors should clarify this point in their revision.

The negative nitrate CI-APi-TOF used in this study is, without the TD-DMA, a fully functional detector for gas phase sulfuric acid, clusters, highly oxygenated molecules, amines, and more (Kürten et al., 2014; Simon et al., 2016; Kirkby et al., 2016). Adding the TD-DMA then provides the bonus of enabling this instrument to measure the particle phase as well without the requirements of cost and space for a second mass spectrometer. Furthermore, when comparing gas and particle phase, it can be done

directly without the need to consider differences in ionization and characteristics of the mass spectrometer as it would be the case when using two separate instruments. While in this work, partitioning studies are not performed yet, they are intended for the future. We thank the referee for pointing out that the ability to measure gas and particle phase with the same instrument deserves more attention, as this is one of the key aspects of our instrument. To highlight this advantage more, and to add also information on the gas phase sampling, page 4, lines 12-32 are modified as follows:

"The coupling between TD-DMA and detector allows for gas phase measurements during particle sample collection periods. This enables an existing gas phase analyzer to measure the particle phase as well without the requirements of cost and space for a second analyzer. Only a small fraction of the gas phase measurement time is lost when it is interrupted by the short evaporation period for the particles. The chemical composition of gas and particle phase can be compared directly without the need to consider instrumental differences in e.g. ionization and characteristics of the mass spectrometer as it would be the case when using two separate instruments. This modular concept allows obtaining a broad picture of both particle and gas phase chemical composition and to observe the condensation, reactive uptake and partitioning of the analyzed substances.

The TD-DMA is a stand-alone instrument that can be attached to any existing technique suitable for real-time chemical analysis of gas phase compounds. In this case, individual compounds relevant for nucleation and early growth of atmospheric nano aerosol particles should be measured. Therefor we used a chemical ionization atmospheric pressure interface time-of-flight (CI-APi-TOF) mass spectrometer with negative nitrate primary ions generated by a corona ion source (Kürten et al., 2011). This technique is specialized for the detection of sulfuric acid, amines and highly oxygenated organic molecules (HOMs) (Kürten et al., 2014; Simon et al., 2016; Kirkby et al., 2016) and characterized for its internal transmission efficiency (Heinritzi et al., 2016) as well as towards its detection efficiency regarding the sulfuric acid concentration (Kürten et al., 2012). For the gas phase measurement, the removal of aerosol particles by e.g. a filter is not required. As the inlet line and ion source are at the same temperature as the analyte aerosol, and the gas composition surrounding the particles is unchanged, the particles do not evaporate significantly on their way to the mass spectrometer and thus

are not detected during gas phase measurements. On the contrary, applying a particle filter would influence the gas phase measurement negatively as a significant fraction of the gas phase analyte, especially sulfuric acid or highly oxygenated organics of low volatility, would adsorb on the filter. "

**4** Some of the language used in the manuscript is imprecise. One example of this is in the discussion of the sampling setup on page 6, lines 1-15. In this section, what does "electrical energy" (page 6, line 5) mean? Does it refer to heat, or voltage? The authors also describe different "parts" of the DMA (page 6, lines 7-9), but don't make use of their labels in the figure, resulting in this section being difficult to follow. In their revision, the authors should carefully read through their manuscript and improve the precision in language.

We agree with the referee that the manuscript benefits from reformulating some passages in the referred section. We therefore reworded the description of the filament on page 6, lines 21-26, as follows:

"The rod carries two thick copper wires with a short piece of platinum wire spot-welded to both of their ends. The electrical resistance of the thin platinum wire is high compared to that of the remaining system. Thus, when a current is applied, a voltage drop occurs mainly across the filament. The deposited energy leads to heating and desorption of the particle material. Electronic relays suitable for high voltage operation are used to disconnect the wires from the current source during collection mode; in this mode a third relay is used to apply a positive high voltage to the wires for electrostatic particle collection."

To make the description of the DMA unit on page 6 easier to follow, we adjusted the labels in figure 2 and the text on pages 6 and 7.

**-** Besides the comments addressed above, three small further correction were made.

(a) On page 8, line 16, a typing mistake was corrected ("reson" -> "reason").

(b) In figure 3, we previously indicated incorrect flow rates, which is now corrected. Since only the labelling in the figure was incorrect, this has no influence on the results of the paper.

(c) On page 13, line 28 we chose the more common expression "dimethylamonium" instead of "dimethylaminium".

[revised manuscript text omitted]

---

## Author Comment (AC2) · 15 Aug 2018

**Reply to Referee 2**

**"Size Resolved Online Chemical Analysis of Nano Aerosol Particles: A Thermal Desorption Differential Mobility Analyzer Coupled to a Chemical Ionization Time Of Flight Mass Spectrometer"**

We would like to thank the reviewer for the comments, which led to an improvement of our manuscript. We addressed the points carefully as described below. Each of the referee's comments is repeated in black font, while the reply is shown in blue font. Changes in the revised manuscript are written in green font. Attached please also find the revised manuscript.

Best regards,
Andrea Wagner and Coauthors

**1** The main issue I see is with a lack of comparison to prior TD-CIMS implementations. The authors do acknowledge the original TD-CIMS work (Voisin et al, 2003), but subsequent development and characterization efforts (Smith et al. 2004, McMurry et al. 2009) are overlooked. I think the authors need to make clearer in this study what is new/unique about their implementation of DMA-thermal desorption-chemical ionisation mass spectrometry. How does their limit of detection compare to prior implementations? I understand that the authors are able to make simultaneous particle and gas phase measurements, but I think this would also be possible with prior implementations of this technique.

The TD-DMA is a modular instrument that can be combined with different detectors. It is an addition to an existing mass spectrometer or other detector, which prepares the particle phase in a way that it can be ionized and analyzed by the detector like the gas phase. To our knowledge, the TDCIMS instrument by Jim Smith and colleagues (Voisin et al., 2003) in contrary chooses a different approach as it exclusively focuses on the

particle phase measurements. It is a full mass spectrometer system including particle phase preparation, ionization and mass spectrometer. Additionally, there are many other instrumental differences between TD-DMA and TDCIMS. These relate to the DMA, the charging unit, the place and procedure of evaporation, etc. However, a detailed comparison between the TDCIMS and the TD-DMA is beyond the scope of our paper. Nevertheless, we do agree that the further developments that have been realized with the TDCIMS should be acknowledged and therefore we have cited the relevant papers in the revised manuscript (Smith et al., 2004 and McMurry et al., 2009). While we agree that the quasi-simultaneous measurement of gas and particle phase chemical composition should in principle be possible with other instruments focusing on nano particle chemical composition measurement, the TD-DMA can still be considered as an innovative concept.

To highlight this, the section describing the benefits of the TD-DMA (Sect. 2 on page 4, lines 9-32) has been rewritten (see also the reply to comment 3 by referee 1):

"The TD-DMA is designed for size resolved chemical analysis of nanometer sized aerosol particles. The particles are charged, a specific size is selected and they are electrostatically collected on a filament. Subsequently, the sampled mass is evaporated in a clean carrier gas to be analyzed by a detector, e.g. a mass spectrometer. In this way, the particle phase is efficiently separated from the gas phase and the concentration is enhanced to meet the detection limit of the analyzer. The coupling between TD-DMA and detector allows for gas phase measurements during particle sample collection periods. This enables an existing gas phase analyzer to measure the particle phase as well without the requirements of cost and space for a second analyzer. Only a small fraction of the gas phase measurement time is lost when it is interrupted by the short evaporation period for the particles. The chemical composition of gas and particle phase can be compared directly without the need to consider instrumental differences in e.g. ionization and characteristics of the mass spectrometer as it would be the case when using two separate instruments. This modular concept allows obtaining a broad picture of both particle and gas phase chemical composition and to observe the condensation, reactive uptake and partitioning of the analyzed substances.

The TD-DMA is a stand-alone instrument that can be attached to any existing technique suitable for real-time chemical analysis of gas phase compounds. In this case, individual compounds relevant for nucleation and early growth of atmospheric nano aerosol particles should be measured. Therefor we used a chemical ionization atmospheric pressure interface time-of-flight (CI-APi-TOF) mass spectrometer with negative nitrate primary ions generated by a corona ion source (Kürten et al., 2011). This technique is specialized for the detection of sulfuric acid, amines and highly oxygenated organic molecules (HOMs) (Kürten et al., 2014; Simon et al., 2016; Kirkby et al., 2016) and characterized for its internal transmission efficiency (Heinritzi et al., 2016) as well as towards its detection efficiency regarding the sulfuric acid concentration (Kürten et al., 2012). For the gas phase measurement, the removal of aerosol particles by e.g. a filter is not required. As the inlet line and ion source are at the same temperature as the analyte aerosol, and the gas composition surrounding the particles is unchanged, the particles do not evaporate significantly on their way to the mass spectrometer and thus are not detected during gas phase measurements. On the contrary, applying a particle filter would influence the gas phase measurement negatively as a significant fraction of the gas phase analyte, especially sulfuric acid or highly oxygenated organics of low volatility, would adsorb on the filter. "

**2** Table 1 and lines 7-27. The table omits techniques where electrospray ionisation is utilized to generate ions. This is an important ionization technique which minimally fragments parent ions, and while it is sparingly used in aerosol science, there are a number of studies demonstrating its potential in aerosol analysis. I would recommend adding electrospray ionisation based methods to Table 1 and mentioning them in the text as these methods can be applied to sub-30 nm particles. Specifically , He et al (2015) showed that nanoparticles can be collected electrostatically and then ions can be generated via electrospray, Horan et al (2017) showed that electrospray-like mass spectra can be collected for aerosol particles without the need for a distinct collection step, and SESI (secondary electrospray ionisation, sometimes called extractive electrospray ionisation), though it cannot distinguish between gas and particle phase, has been employed in several studies (Doezema et al. 2012, Gallimore and Kalberer 2013).

We thank the referee for making us aware of these exciting new techniques using electrospray ionization. As this manuscript focuses on the chemical analysis of sub 30 nm particles for new particle formation, we find the approaches by He et al., 2015 and Horan et al., 2017 to fit in well and added them to Table (1) and the overview text on page 3, lines 26 - 31:

"A new technique for the discontinuous particle phase analysis is the Eletrostatical Precipitation Electrospray Mass Spectrometer (EP-ESI-MS) by He et al. (2015). Particles are charged and electrostatically collected. Thereafter, the material is brought to the gas phase and softly ionized at the same time by using the collection surface as an electrospray tip. The Droplet Assisted Inlet Ionization (DAII) (Horan et al., 2017) condenses water on particles and rapidly evaporates them in a heated inlet. It is a continuous method and has proven the ability to measure test particles as small as 13 nm. "

We did minor adjustments to the distinguishing criteria in order to include the new methods (page 3, line 8-12):

"They can be distinguished based on the following main criteria (see Table 1): (1) a discontinuous principle to enrich the analyte vs continuous measurements,  (2) size resolved methods vs integral sampling methods, (3) particle evaporation method, e.g. by thermal desorption or laser evaporation, and  (4) ability to analyze gas and particle phase vs only particle phase. "

**3** Equation (1) and the surrounding analysis. While the authors do note "Note that the aerosol coming from the DMA is not strictly monodisperse; instead the DMA provides a Gaussian-shaped size distribution," it does not appear they account for this in their analysis or explain to readers why they do not need to account for this. Equation (1) is not the true transfer function of the DMA; this would only be the transfer function if the first DMA had extremely high resolution relative to the test DMA. Looking at the sheath flow rate employed in the TD-DMA relative to the aerosol flowrate, this is probably the case and the authors' analysis is ultimately fine, but I would advise the authors to show

this more clearly. Specifically, the number concentration of particles at the outlet of the first DMA is given by the equation:

$$N_2 = \int_0^\infty \frac{dn}{dd_p} \tau_G\left(d_p\right) dd_p$$

where $\tau_G(d_p)$ is the Grimm DMA transfer function/transmission function. For particles exiting the TD-DMA, the number concentration is:

$$N_1 = \int_0^\infty \frac{dn}{dd_p} \tau_G\left(d_p\right) \tau\left(d_p\right) dd_p$$

The ratio $N_1/N_2$ is hence not given by equation (1), but is:

$$\frac{N_1}{N_2} = \frac{\int_0^\infty \frac{dn}{dd_p} \tau_G\left(d_p\right) \tau\left(d_p\right) dd_p}{\int_0^\infty \frac{dn}{dd_p} \tau_G\left(d_p\right) dd_p}$$

A common (reasonable) assumption is that $\frac{dn}{dd_p}$ is a constant over the region where $\tau_G\left(d_p\right)$ takes non-zero values. This leads to:

$$\frac{N_1}{N_2} = \frac{\int_0^\infty \frac{dn}{dd_p} \tau_G(d_p) \tau(d_p) dd_p}{\int_0^\infty \frac{dn}{dd_p} \tau_G(d_p) dd_p} \neq \tau\left(d_p\right)$$

Therefore, the method noted in the manuscript at present does not yield $\tau\left(d_p\right)$ unless $\tau_G\left(d_p\right)$ is significantly narrower (higher resolution) than $\tau\left(d_p\right)$. At an aerosol to sheath flow ratio of 3/5, I assume the assumption is reasonable, but does need to be justified or better yet, explicitly shown in the manuscript.

The referee is correct, formula (1) on page 7 is only valid in a tandem DMA system if the resolution of the first DMA is much higher than that of the second. We thank the referee for pointing this out. We followed the suggestion and validated our approach by an example calculation:

A typical particle size selected in the TD-DMA is 15 nm. Based on the functions provided we calculated the value of

$$\frac{N_1}{N_2} = \frac{\int_0^\infty \tau_{Grimm,15\ nm}(d_p) \cdot \tau_{TD-DMA,15\ nm}(d_p) \cdot dd_p}{\int_0^\infty \tau_{Grimm,15\ nm}(d_p)\ dd_p}$$

using the theoretical transfer functions ($\tau$, also sometimes termed $\Omega$), including diffusion, from Stolzenburg and McMurry, 2008. The resulting value for $\frac{N_1}{N_2}$ was then compared with the approximation $\tau_{TD-DMA,15nm}(15\ nm)$ (i.e., the maximum value of the transmission curve for the selection of 15 nm particles). The comparison between the two approaches (exact and approximated) yields a difference of only 7.8 %. Therefore, we consider the approach to handle the aerosol coming from the first DMA as quasi-monodisperse as valid for the given instruments and flowrates

On page 8, lines 3-6, it is now mentioned that equation (1) is only an approximation, which is valid for the present study. We do think, however, that it is not necessary to include further equations.

"Note that the aerosol coming from the first DMA is not strictly monodisperse; instead the DMA provides a Gaussian-shaped size distribution. As in this case, the resolution of the first DMA is much higher than that of the second DMA, equation (1) introduces only a small error; using the methods by Stolzenburg and McMurry (2008) this can be demonstrated for the relevant sizes of the TD-DMA. "

- Besides the comments addressed above, three small further correction were made.

(a) On page 8, line 16, a typing mistake was corrected ("reson" -> "reason").

(b) In figure 3, we previously indicated incorrect flow rates, which is now corrected. Since only the labelling in the figure was incorrect, this has no influence on the results of the paper.

(c) On page 13, line 28 we chose the more common expression "dimethylamonium" instead of "dimethylaminium".

**References**

[revised manuscript text omitted]

---

## Author Response (AR2)

**Reply to Associate Editor report**

Dear Johannes Schneider,

Thank you for this useful comment.
The comment is repeated here in black font; our reply is written in blue font. Changes in the revised manuscript are written in green font. Attached please also find the revised manuscript, and the revised manuscript with highlighted modifications.

Best regards,
Andrea Wagner and Coauthors

Associate Editor Decision: Publish subject to minor revisions (review by editor) (22 Aug 2018) by Johannes Schneider
Comments to the Author:

Dear Andrea Wagner,
thank you very much for the revisions which improved the manuscript significantly. I will be happy to accept it for publication in AMT after consideration of the following point:

On page 16, you discuss the effect of double-charged particles. There is a discrepancy between the original text and the new addition: the statement of "two times the diameter and eight times the mass", because in your addition you correctly state that the doubly-charged particles with same mobility as a 15 nm particle is 21 nm in diameter, i.e. not 30 nm. Additionally, I can not reproduce the result that "only a fraction of 0.009 of the particle mass in the sample would originate from doubly charged larger particles". Apparently I missed some steps of the calculation. Please explain in more detail.

Best regards,
Johannes Schneider

This is correct. Assuming the doubly charged particles would have twice the diameter, neglects the Cunningham slip correction factor for the conversion of electrical mobility to diameter, which is not appropriate for these small sizes. We reworded the according sentence. To make the calculation more comprehensible, we now describe it in more detail. The correspondent paragraph on page 15, line 28 to page 6, line 9 now reads:

[revised manuscript text omitted]